# Torn Between Identities: A Hermeneutic Phenomenological Study of Nurses’ Dual Allegiance During COVID-19 and Armed Conflict

**DOI:** 10.3390/nursrep16010012

**Published:** 2025-12-31

**Authors:** Nurit Zusman, Caryn Scheinberg Andrews

**Affiliations:** Henrietta Szold Hadassah School of Nursing, Faculty of Medicine, Hebrew University, Ein Kerem, Jerusalem 9112102, Israel; candrews613@gmail.com

**Keywords:** moral distress, virtue ethics, armed conflict, identity, nurses, COVID-19

## Abstract

**Background/Objectives**: While nurses showed a willingness to work during the pandemic and wartime, little is understood about how they managed the conflict between their roles as caregivers and personal or family obligations. They are deemed “essential workers,” risking their safety to fulfill their duties. **Objectives**: This study aims to explore the lived experience of nurses during COVID-19 and wartime, delving deeper into their emotional and moral experiences, providing insights for nurses and nursing management about how nurses negotiate dilemmas. **Methods**: A focused interpretive, hermeneutic, phenomenological approach was employed. From December 2022 to January 2023, ten hospital-based nurses from two hospitals were purposively sampled for in-depth, semi-structured interviews, which were transcribed and analyzed. The study was approved by the University Ethics Committee (31102022). **Results**: The essence of “ Moral Conflicts of Dual Identity and Dual Allegiance” revealed profound moral and emotional struggles among nurses. Four key themes emerged: (1) Moral Stressors and Identity Negotiation, (2) Competing Responsibilities and Ethical Double-binds, (3) Virtual and Practical Wisdom in Crises, (4) Responses of Stress and Erosion of Support **Conclusions**: Understanding nurses’ ethical dilemmas is essential for healthcare leadership. Leaders must make it a priority for workplace safety for their nurses. In wartime, it is not obvious that the workplace is unsafe; leaders must foster open dialog and support systems in response to these crises. This study highlights the significance of peer support, emphasizing the need for policies that address the complex moral challenges nurses face daily.

## 1. Introduction

Between 10 and 22 May 2021, during a 12-day conflict and the ongoing COVID-19 pandemic, nurses in southern Israel faced extraordinary challenges [1]. In cities such as Ashkelon, frequent rocket attacks and pandemic restrictions forced nurses to navigate competing demands between personal and family safety and their professional responsibilities. Nurses often had to decide whether to leave their families, who were themselves at risk, in order to care for patients with COVID-19 and those injured in the conflict.

More recently, the large-scale attack of 7 October 2023 [2], and, since June 2025 [3], escalating regional tensions—including missile fire from Iran and Yemen—have continued to threaten healthcare settings. These events highlight the persistent dual vulnerability of nurses, who must provide urgent care while simultaneously managing threats to their own and their families’ safety [3].

Although the psychological toll of the COVID-19 pandemic [4] and the strain of working in conflict zones have each been well documented [5,6], the *combined* impact of concurrent pandemic and war conditions remains underexplored. In particular, little is known about how nurses negotiate dilemmas between professional duty and personal safety when both are acutely in jeopardy.

Focused interpretive hermeneutics [7] is an effective method for several reasons, as it captures the complexity of individuals’ subjective experiences in challenging situations. By considering the socio-cultural context and elevating the voices of marginalized populations, such as nurses in crises, provides deeper insights into decision-making, dilemmas, and emotional responses. This interview approach integrates various dimensions of human experience, facilitating a comprehensive understanding of the interconnectedness of these aspects during times of conflict and trauma [7].

This study addresses this gap by examining the lived experiences of nurses working under overlapping missile-attack and pandemic conditions, offering new insight into moral stress, resilience, and decision-making in high-risk environments. Our research gives voice to a frequently overlooked population, vulnerable nurses, whose lived realities have broader implications for global health, trauma-informed leadership, and occupational safety [7].

### Background

Research on the effects of COVID-19 on nurses is extensive. Nursing was reportedly strengthened by COVID-19 by the broadening of nursing responsibilities [4] to previously restricted subspecialties [8,9]. Nurses worked in different units and learned new skills, such as those in intensive care units [8]. Another aspect of nurse’s response to the pandemic was studied in Sweden [8] whereby researchers explored the notion of altruism with nurses caring for COVID-19 patients despite putting their own lives at risk. In another study of 448 Jordanian nurses, who were mostly female, researchers compared acute stress disorder (ASD) as an aspect of psychological distress among the population [10]. While nurses experienced substantial levels of stress, anxiety, and depression during the COVID-19 pandemic, many also demonstrated remarkable resilience, continuing to fulfill their professional responsibilities related to self-efficacy and good coping skills [10]. 

Nursing work in conflict zones has likewise been studied. Following the war “Operation Guardian of the Wall”, researchers explored the feelings of “sense of danger” or “reporting to work” during wartime [11]. Among 390 participants (77% female) in southern Israel, those older and those in management positions missed work less, and missing work was related to an increased sense of danger, the need for childcare, and an impact on mental health [11]. 

A broader regional study assessing perceived threat and willingness to work during terrorism compared nurses’ responses to wartime and pandemic scenarios [12]. Among 249 largely female nurses across four hospitals, most were **unwilling** to report to work during a terrorist threat, yet 92% stated they **would work** during COVID-19 and believed almost all colleagues would as well [12].

Despite robust literature on each crisis independently, limited research explores the *combined* effects of simultaneous pandemic and wartime conditions. Little is known about how nurses navigate and cope with the dual challenges of working during a pandemic while also dealing with the stresses of wartime conditions. Limited research exists about overlapping stressors that affect nurses’ mental health, coping mechanisms, and safety dilemmas. The absence of evidence on these overlapping realities restricts development of targeted organizational strategies for nurses working under dual crises.

The purpose of this study was to understand the lived-experiences of nurses during the pandemic, while simultaneously living under wartime conditions, using a caring inquiry approach grounded in hermeneutic phenomenology [7]. Understanding the lived experience will help organizations, managers, and peers provide caring support. The primary research question was: *What is the meaning of the lived experience of nurses during a war, yet still facing the dilemma of going to work in a hospital caring for patients with COVID?* The secondary question was, *What organizational support do nurses perceive that they need from the organization to help endure such conditions?* This inquiry contributes to the knowledge of frontline nursing in the contexts of sustained national and occupational stress.

## 2. Research Design

### 2.1. Methods

We used a focused interpretative (hermeneutic) phenomenology approach (philsophical lens) perspective inspired by Heidegger [13] and Gadamer [14] to understand the lived experience of nurses in a war zone who also worked as nurses in local hospitals during the COVID pandemic, to understand their dilemma of choosing whether to go to work or stay home with their families. This approach views understanding as a dialogical and reflective process [15] in which both participant and researcher contribute to meaning-making [15,16], allowing the analysis to move beyond description toward uncovering the deeper significance of participants’ experiences. Interpretive phenomenology (IPA) [16,17,18] was used to operationalize the understanding of the “essence” of the lived experiences of individuals For the method of analysis we used IPA to operationalize the interpretation, thus using a blended approach.

### 2.2. Setting and Sample

Purposive sampling, snow-ball recruitment was used to recruit ten hospital-based nurses from one hospital in the south and one in the center of the country (see Table 1). The inclusion criteria were hospital-based nurses during COVID-19 and wartime. Emails were sent to the nurses in the first hospital inviting them to participate, and they responded to the PI that they wanted to participate. In the second hospital, nurses were introduced to the PI by fellow nurses. The invited nurses called the PI and confirmed participation. There were no incentives for participation were offered. There were no nurses that refused to participate or dropped out.

### 2.3. Data Collection

For the present research study, following ethics approval, the participants were consented and interviewed from December 2022 until January 2023.

The PI, known to some of the participants as the “Director of Nursing,” was aware that it was possible that some answers were connected to her relationship as their manager. Data were transcribed and translated from Hebrew into English by a professional transcriptionist/certified translator. The two authors reflected on the interviews data, right after the interviews by a debriefing process. The interviews were analyzed iteratively in both English and Hebrew. The researchers went back and forth in both languages to make sure they understood what the participant was saying. Both are female, PhD researchers. Both are employed by the University School of Nursing. The duration of the interviews ranged from 30 to 60 min. Data collection continued until thematic saturation or interpretive completeness was reached. “After the ninth and tenth interviews, no new themes emerged and existing ones were consistently confirmed”.

The interview questions were initially informed by issues raised during meetings held when the researcher (PI) served as the hospital’s “Director of Nursing”, aimed at enhancing the quality of care and supporting nursing staff. They were open-ended and the researcher asked for clarification and probing, such as “Can you tell me about a moment when….”. These questions were subsequently refined and aligned with the study’s method. Additional adjustments were made following the first round of interviews to improve clarity and relevance. The questions are found in Table 2.

### 2.4. Data Analysis

Data analysis was guided by a hermeneutic phenomenological approach inspired by Heidegger and Gadamer [12,13], emphasizing interpretive understanding [15,16,17], the role of researcher pre-understandings, and movement between parts and whole through the hermeneutic circle. The goal was uncovering meaning structures [18] underlying experiences of personal and professional conflict. Throughout the analysis, researchers engaged in reflexive discussions to examine how their own professional backgrounds and emotional responses influenced interpretation.

### 2.5. Ethical Consideration and Trustworthiness

Prior to beginning data collection, the study was approved by the University Ethics Committee (Study # 31102022). All procedures were followed according to the regulations of Helsinki Declaration principles. Verbal informed consent was obtained from all participants, clearly explaining study purposes, voluntary participation, confidentiality protections, and withdrawal rights without consequences.

PI (1), an experienced qualitative interviewer, verbally obtained consent from participants prior to the interview. The interviews took place over zoom, verbal consent was obtained, on zoom and recorded, the camera was closed, the participants were labeled by number with no identifying information recorded, and no one else was present besides the participant and researcher PI (1).

### 2.6. Researcher Characteristics and Reflexivity

All transcripts were reviewed in Hebrew and English iteratively by both researchers. PI (1) (native Hebrew speaker) and Co-PI (native English speaker) first engaged in independent analysis, then compared meaning units and interpretive possibilities. Given that both researchers experienced the COVID-19 pandemic and concurrent armed conflict alongside participants—and that the PI (1) holds senior clinical and academic leadership roles—heightened reflexivity was intentionally embedded throughout the analytic process to mitigate potential interpretive bias. An outsider researcher confirmed data consistency with team interpretation.

To manage proximity and professional influence, analytic decisions were grounded in participants’ verbatim accounts, with explicit attention to distinguishing participants’ meanings from researchers’ professional or leadership perspectives. The PI (1) actively monitored moments of resonance with her own experience and deliberately deferred interpretive closure until meanings were examined collaboratively. Interpretations were not finalized unilaterally; rather, they were subjected to dialogical review, questioning, and return to the original text.

Analysis involved line-by-line engagement attending to nuances, metaphors, emotional tones, and contextual cues signaling deeper experiential meaning. Working iteratively through the hermeneutic circle, we moved between individual statements, full transcripts, and the dataset as a whole. Emergent experiential themes were identified and clustered into broader interpretive categories through cycles of reinterpretation and comparison across cases.

The epistemological assumption underlying this study was that the PI (1), who shared the lived experience with the participants, had an insider perspective. While this proximity allowed the PI (1) to deeply understand the context, she had to “bracket” her own experiences to avoid bias during analysis. The dual-perspective interpretive team enhanced credibility: PI contributed contextual insider insight while Co-PI provided essential reflexive outsider lens. Their dialogical engagement, questioning assumptions and returning repeatedly to text, strengthened interpretive depth.

### 2.7. Rigor and Trustworthiness

Prolonged fieldwork over one year enabled deep understanding. Member checking involved re-interviewing all first-set participants. Triangulation utilized another researcher reviewing data and results. Rich descriptions and diverse perspectives enhanced transferability. Audit trail (shared Excel file with transcripts and video tapes) and peer debriefing ensured dependability. Reflexive journaling minimized biases.

## 3. Results

### 3.1. Participants

Participants were predominantly female nurses, aged 33–34 years, married, with all having bachelor’s degrees and half holding master’s degree. (see Table 1).

### 3.2. Core Meaning Structure

Analysis uncovered a core meaning structure that shaped participants’ experiences of navigating the dual situation of caring for COVID patients and simultaneously working under wartime conditions, which emerged from the data as, **“Moral Conflict of Dual Identity and Dual Allegiance.”** This structure reflects the persistent tension they described between their professional responsibilities as caregivers/nurses and their personal identities as mothers, spouses, and family members. Participants did not experience this as a single moral decision but as an **ongoing, embodied process of negotiating and interpreting competing obligations** amid rapidly shifting circumstances. The assumption was that the experiences of nurses in high-risk environments, particularly during simultaneous conflicts and pandemics, significantly affect their decision-making, professional identities, and emotional well-being, highlighting the need for a deeper understanding of the moral and ethical dilemmas they encounter.

Participants consistently articulated how moral stressors, situational complexity, and the constant threat of violence during both the COVID-19 pandemic and the concurrent war intensified this tension. They described repeatedly weighing the risks to their families against their duty to provide care, often moving between feelings of fear, responsibility, guilt, and resolve. This dynamic process shaped how they understood “what the right thing to do” meant in each moment.

Across interviews, researchers found that participants made sense of this conflict through **virtue-based moral reasoning**, drawing on courage, compassion, respect, honesty, and practical wisdom to guide decisions. The dilemma of “being a nurse” versus “being a mother” emerged as a recurrent expression of this dual identity and therefore, dual allegiance struggle, encapsulating the emotional and ethical complexity of working within overlapping crises. These virtues were not abstract ideals but lived orientations that nurses relied on to navigate ethical stress and to reconcile their professional and personal identities. Together, these findings illustrate how nurses interpreted and constructed meaning in the face of profound moral disruption, **revealing virtue ethics as the interpretive thread** through which participants understood and enacted their moral agency.

Through a hermeneutic–interpretive analysis, grounded in phenomenology, and carried out using IPA procedures, four overarching structures of meaning were uncovered. These themes reflect the interpretive processes through which participants made sense of their lived experiences amid the intersecting crises of COVID-19 and war: (1) Moral Stressors and Identity Negotiation; (2) Competing Responsibilities and Ethical Double-Binds; (3) Virtue and Practical Wisdom in Crises; and (4) Responses to Stress and Erosion of Support.


**Theme 1: Moral Stressors and Identity Negotiation**


Participants encountered ethically complex situations requiring moral discernment within contexts of violence and instability. These were deeply affective moments where nurses felt “torn” or “heavy,” negotiating professional and personal duties within compromised safety. Their identity negotiation/conflict was described as a shifting sense of self, at times identifying more with a professional role, at others with a familial one. This emerged not only in action, but in being, who one is called to be in a particular moment.

Communication barriers and emotional isolation compounded distress:

“*We did not talk about that, about the war. Every person kept to themselves and we did not talk about the escalation in real time. It was like walking on eggshells*”(P1, Internal, 42F)

“*I felt that no one truly understood what we went through and no one really cared.… It’s like they used us because they needed us, they put us in very complicated units, without proper instructions or preparation*”(P4, ICU, 33F)

Personal impact of witnessing violence created lasting psychological effects:

“*During the war it was a little bit more complicated for me, especially because a missile entered a home, in a shelter room and a little kid was murdered, and it gave me anxiety… it’s not always in my control and things can happen*” (P6, Onc, 38F)

Professional calling vs. parental responsibility: Despite profound conflict, nurses drew upon a deep sense of purpose and meaning, a deep sense of calling:

“*Knowing I must go on, for my children, and know life goes on after the situation*”(P1, IM, 42F)

“*Simple. It’s a calling. This satisfaction. I have always known that I want to work in this profession… I feel that I’m at the right place*”(P2, IM, 38F)

Yet leaving family during dangerous situations created significant strain, with one nurse cutting short maternity leave:

“*I myself shortened my maternity leave to come back and help at work… I came back in August, it was something that was unthought of*”(P3, IM, 39F)


**Theme 2: Competing Responsibilities and Ethical Double-Binds**


Participants’ descriptions of ethical double-binds disclose a meaning structure of “being-responsible under threat.” Fulfilling one obligation (to patients, colleagues, the system) was experienced as simultaneously exposing another vulnerability (family, self, home), so that every action was saturated with moral consequence. This tension was not merely cognitive but “lived in the body” as fatigue, guilt, and a persistent inner unease, revealing how ethical conflict settled into participants’ embodied being-in-the-world.

Daily commuting became a source of constant anxiety. For example, travel emerged as more than physical travel between home and work; it became a “liminal space of exposure and risk”, where the nurse is pulled between two moral worlds:

“*Going back and forth from home to work, and then home again make me feel exposed and vulnerable… maybe something would happen to me, God forbid something may happen at home, maybe a rocket would fall nearby…*”(P1, IM, 42F)

Movement through space is interpreted through a horizon of danger and responsibility, with the nurse imaginatively holding both self and family within the same field of threat. In this context, every choice or decision required constant vigilance and moral calculation, Another example, what would happen to the family if the nurse was injured while traveling? Participants were not simply “making choices,” but continuously interpreting and recalibrating their moral priorities within unstable conditions. The ethical double-bind thus appears as a lived structure of ongoing moral calculation, where nurses inhabit a world in which *any* choice may feel like a partial betrayal of another responsibility.

“*Everything you do you need to be careful, you need to think about the alternatives of each decision, and the priorities etc…*”(P9, NEUSURG, 44M)


**Theme 3: Virtues and Practical Wisdom in Crisis**


Virtues and practical wisdom function as *meaning structures* and modes of being, not just coping skills. Virtues are considered as existential anchors—ways of being that enable nurses to remain morally grounded amid uncertainty, threat, and emotional strain, and a way of supporting themselves. It was the way the nurses made sense of themselves.

**Internal virtues as anchoring:** Nurses drew upon resilience, self-compassion, and reflective practice, constituting a form of self-support allowing them to endure and make meaning.

“*I developed the capacity to understand and truly see the person in front of me. …I truly realized that there was a way to have an impact on people if we know how to reach out to them.*”(P4, ICU, 33F)

Professional virtues such as, courage, compassion, integrity, respect, and honesty, emerged as orienting structures of meaning in decision-making. These were lived realities cultivated over time:

“*It starts and ends with human contact, with caring. It’s something I have learned in time*”(P3, IM, 39F)

“*To be very gentle and careful with people. In life as well in general*”(P9, NEUSUR, 44M)

**Phronesis (practical wisdom):** Participants demonstrated practical moral wisdom rooted in experience and relational knowing. Decisions were guided by nuanced understanding of context, relationships, and human condition:

“*I developed the capacity to understand and truly see the person in front of me… I realized there was a way to have an impact on people if we know how to reach out to them*”(P4, ICU, 33F)

This wisdom included recognizing personal limitations:

“*I am not made of steel, I am made of porcelain, it’s a mistake to think that you are strong. You need to be very cautious when you decide that you’re strong. Because you might get yourself into situations that would be very hard to get out of…*”(P9, ICU, 44F)


**Theme 4: Responses to Stress and Erosion of Support**


Participants’ responses to stress reveal shifting **modes of being-in-the-world** under prolonged moral pressure. Adaptive and maladaptive reactions were not simply coping strategies but *interpretive orientations*—ways of understanding themselves, others, and the moral landscape they inhabited. These narratives reflect a meaning structure of fractured moral community, in which nurses oscillated between moments of connection and profound abandonment. Their responses reveal not just stress reactions but interpretive efforts to maintain moral integrity and relational grounding in a world that had become unpredictable, isolating, and ethically demanding.

**Adaptive vs. maladaptive responses:** Adaptive responses included intentional reflection, boundary-setting, and dialog with peers. Maladaptive responses included emotional detachment, self-doubt, and burnout. These revealed deeper modes of being-in-world under moral pressure.

Finding meaning in patient connections sustained resilience:

“*In the end of the day, I know that behind any person there is a family. And as I see their appreciation and gratitude reassures me and gives me more energy to carry on*”(P1, IM, 42F)

However, constant vigilance created heightened sensitivity:

“*Even the smallest thing I hear over the news makes my body go on defensive mode, and you tell yourself it starts again*”(P1, IM, 42F)

Collegial relationships functioned as vital spaces of shared understanding and moral affirmation. Informal peer support—often spontaneous and relational—temporarily restored a sense of communal grounding:

“*For example, one co-worker of mine, you could say she was in charge of the staff’s morale, which helped a lot. I mean, also when she was inside, she would suddenly start dancing, would take a patient and do something with it. Whether she was inside or outside, it helped a lot*”(P2, IM, 38F)

Yet a significant gap was described as a profound erosion of institutional and organizational support, existed between the moral need for support and what was available:

“*No one is really there for us, to listen to us and help us process*”(P3, IM, 39F)

Prolonged crisis eroded team cohesion:

“*I think that for the staff, in those 2 years, we lost the sense of ‘togetherness’ that we had developed prior to the COVID. Maybe it happened because of the quarantines that were so long. Each one of us was at home, surrounded only by our own problems and solutions…*”(P4, ICU, 33F)


**Interpretive Synthesis**


Together, these themes reflect the hermeneutic nature of the participants ethical experience: a continuous process of interpreting and reinterpreting through the lens of identity, relationships, and vulnerability. The central meaning structure of “Moral Conflicts of Dual Identity and Dual Allegiance” emerged not as singular event but as living tension revealing vulnerability and moral depth of human experience in caregiving professions. Ethical decision-making emerged as a way of being grounded in virtues and practical wisdom rather than discrete choices. This ongoing moral navigation highlights the complexity and humanity of caregiving in the intersecting context of war and the pandemic.

For **Research question 1**, the findings show that nurses who are also parents experienced persistent moral tension, heightened vulnerability across personal and professional spheres, and challenges reconciling competing obligations under chronic uncertainty—adding depth to pandemic research that tends to emphasize constraints, workload, and psychological outcomes [19].

For **Research question 2**, participants identified clear organizational needs, including transparent communication, flexible scheduling, peer support, and acknowledgment of family responsibilities. These align with recent evidence linking organizational support, ethical climate, and moral resilience to reduced moral distress and improved workforce outcomes [20].

## 4. Discussion

This study sheds light on the profound ethical, emotional, and existential challenges faced by the participants/nurses, some of whom are also parents, during the COVID-19 pandemic and periods of armed conflict. Prior research has shown that the pandemic produced high levels of moral distress, ethical uncertainty, and emotional strain among nurses, often linked to compromised standards of care, role conflict, and chronic exposure to suffering [21,22,23].

Using a hermeneutic-phenomenological approach, this study extends prior work by showing how nurses experience moral distress as an ongoing condition embedded in daily life, rather than an isolated ethical dilemma [7]. The “**Moral Conflicts of Dual Identity and Dual Allegiance**” are experienced as persistent negotiations between the responsibilities of caregiving and personhood, intensified by the constant threat of personal and familial harm. This aligns with existing literature on moral distress in nursing but extends it by highlighting the existential dimension of identity conflict, particularly in settings marked by acute danger and uncertainty.

### 4.1. Why the Dual-Crisis Context Reveals New Aspects of Moral Distress

The combination of the COVID-19 pandemic and armed conflict creates an unprecedented ethical environment, increasing and distorting nurses’ moral distress in ways that have not been previously examined. Prior studies typically examine moral distress within a single crisis—either a public health emergency [24] or a security threat—but not both simultaneously. In addition, while dual-role tensions and moral injury have been examined among military and defense-force nurses [25] yet they have not been fully conceptualized for civilian nurses working in overlapping crises.

In this study, nurses navigated *layered risks and competing dangers*, where high-risk clinical duties coexisted with direct threats to their families’ safety. This dual exposure produces a qualitatively distinct form of moral conflict: nurses experience constraints on patient care at the same time that they face existential concerns about protecting their own children, partners, and homes. These overlapping crises blur the line between personal vulnerability and professional obligation, intensifying the sense of “dual allegiance” and generating moral tensions grounded not only in external constraints but also in deeper questions of loyalty, identity, and moral responsibility. This unique context identifies **compounded moral distress**—rooted in simultaneous caregiving roles, heightened danger, and chronic uncertainty, which has not been fully examined in previous research.

### 4.2. Theoretical Framework: Virtue Ethics

From a hermeneutic phenomenological perspective, moral distress in dual-crisis conditions is understood as a disruption in nurses’ lived world—a blurring of boundaries between home and hospital, safety and danger, and care for self versus care for others. This interpretation aligns with recent phenomenological findings showing that nurses understand their moral experience not only in terms of workload or resources but through who they are, what they value, and how they make meaning of crisis situations [20,26,27].

Virtue ethics provides a useful framework for interpreting how nurses navigate such dilemmas, emphasizing the cultivation of character traits such as courage, compassion, and “phronesis”(practical wisdom), as guides for nurses when they lack external guidance on rough decisions or choices. No existing guidelines exist, and nurses must rely on their own values” [27] (p. 3447). Nurses who face the decision to go to work, taking care of COVID-19 patients versus the need to stay home to protect their families from danger, face internal decisions are grounded in this principle of phronesis or practical wisdom.

As an ethical approach, virtue ethics shifts the attention to personal virtues that guide moral judgment and action, rather than strict adherence to rules or consequences. For example, in the context of the COVID pandemic, practitioners drew on inner resources and professional discretion, displaying virtues such as professional wisdom, care, respectfulness, and courage as they navigated complex situations [8]. Although there is research on nurses’ “willingness to work during the pandemic”, and “responding to human need” from an altruistic perspective, there is little data about the nurses deciding whether home and family are put aside for the “greater good” [12].

The study also elucidates the importance of the organization as a broader “ethic of care.” When organizations demonstrate care for the participants’ safety, emotional needs, and family responsibilities, they strengthen the capacity to act compassionately and ethically toward patients and families [14]. This reciprocal relationship underscores that virtue-based practice is not only shaped by the individual’s character but also by the moral climate created by the organization [20].

### 4.3. Moral Distress Amidst Dual Crises

Nurses have faced significant moral distress during the COVID-19 pandemic, often stemming from institutional constraints that impede their ability to provide optimal care. This distress is exacerbated when personal and professional roles collide, as seen in nurse-mothers balancing patient care with family responsibilities. The moral distress experienced by nurses has been linked to factors such as the volume of care for infected patients, access to personal protective equipment, and communication from leadership, all of which have longer-term mental health implications [19].

The tension between professional obligations and personal responsibilities is a recurring theme. Nurses grapple with decisions that test their professional integrity and personal conduct, such as prioritizing patient care over personal and or family needs during crises. This internal conflict underscores the recognition of the importance of a virtue ethics framework, which supports moral resilience and integrity in the face of such challenges. The pandemic has provided an opportunity to explore the links between moral distress, moral resilience, and the emergence of mental health symptoms in healthcare workers, highlighting the importance of supporting nurses in maintaining their professional integrity [20].

Organizational support, including transparent communication and access to resources, empowers nurses to exercise sound moral judgment [28]. Peer support and self-care practices also contribute to resilience, enabling nurses to navigate moral complexities effectively. Leaders are encouraged to communicate transparently to decrease nurses’ moral distress and the negative effects of global crises on nurses’ longer-term physical and mental health [19].

### 4.4. Implications for Practice and Policy

Understanding the participants’ dilemmas in general, and especially during a crisis, is important for management and leadership. Managers and leaders need to actively ensure a safety workplace for nurses before, during and after a crisis. They also need to understand the situation and initiate dialog with staff nurses, thus imparting a sense of caring and partnership. Nurse managers are encouraged to focus on their staff as “partners rather than subordinates” [29] (p. 238), and as unique contributors to the environment. The results of this study support this need as expressed in the dialogs.

The dual challenges of the COVID-19 pandemic and regional conflict have amplified the moral and emotional burdens on nurses and family members. A pathway to navigate these complexities, fostering moral resilience and integrity, with the necessity and obligation of the organization to offer “protection and safety to the working nurse force is of paramount importance. By prioritizing the cultivation of personal virtues [9] and robust support systems, the nursing profession can better prepare for future crises, ensuring the well-being of both healthcare providers and patients. Nurses will show up for work [12], research has shown that, and this study supports that. The price of “health and safety” is large however, and the onus of support lies on the organization and healthcare system to support the backbone of healthcare, not to risk their safety in lieu of their internal “call” for nursing. Organizational targeted education should be introduced to discuss the values of virtue ethics spoken openly as part of self-knowledge and humanity.

The findings highlight that in dual-crisis settings, where pandemic conditions overlap with active armed conflict, many common organizational recommendations become impractical or unsafe. Participants’ experiences show that meaningful support must reflect the realities of unpredictable alerts, unsafe travel routes, and rapid shifts from routine care to personal danger. In these conditions, organizations can strengthen resilience by providing only those interventions that remain feasible during acute threats, such as real-time, standardized communication during alerts; clear guidance on when staff should delay travel or shelter in place; protected internal shelter spaces within hospitals; and rapid shift-change coordination that minimizes unnecessary commuting during peak danger. Emotional and peer support should be offered through brief, structured check-ins integrated into existing workflows rather than adding meetings that require travel.

### 4.5. Limitations

This study has several limitations that should be considered when interpreting the findings. Consistent with the qualitative and contextual nature of hermeneutic phenomenological research, the findings are not intended to be statistically generalizable. First, the sample was drawn from a specific region experiencing simultaneous pandemic conditions and active missile fire; therefore, the experiences described here may differ from those of participants in other geographic, cultural, or organizational contexts. Although the dual-crisis setting is central to the study’s contribution, it naturally limits direct transferability to settings where health and security crises occur independently or under less acute threat.

Second, as a hermeneutic phenomenological study, the findings reflect an interpretive engagement between participants and researchers. The research team was living through the same crises at the time of data collection and analysis, and this shared exposure may have influenced the interpretive lens through which experiences were understood. **The strategies used to address researcher positionality and professional influence are detailed in the Reflexivity subsection of the Methods.** While such positionality is inherent to hermeneutic inquiry—and can enhance depth of understanding—it may also introduce bias by shaping the questions asked, the meanings emphasized, or the resonance of particular themes.

Third, reliance on voluntary participation may have led to underrepresentation of individuals who were unable or unwilling to speak about their experiences due to emotional fatigue, trauma, or time constraints during the crises. Finally, the rapidly evolving nature of both the pandemic and the conflict meant that interviews captured experiences at specific points in time. Moral challenges and coping strategies may shift as circumstances stabilize or intensify, and longitudinal trajectories were beyond the scope of the study.

**Despite these limitations, the use of rich, contextualized descriptions allows for analytic transferability, enabling readers to consider the relevance of the findings to other high-stress, ethically complex, or crisis-affected healthcare settings.** The study provides rare and timely insight into moral distress during overlapping crises, offering a foundational understanding of the ethical and emotional realities faced by healthcare personnel in high-risk environments.

## 5. Conclusions

This research is critically important because it addresses a significant gap in understanding how nurses navigate ethical, emotional, and practical issues during compounded crises—specifically, the simultaneous experience of the COVID-19 pandemic and wartime conflict. While much has been written about the impact of either pandemics or war on healthcare workers individually, little is known about how these overlapping stressors affect nurses’ ability to fulfill their professional responsibilities while protecting their families and managing personal risk. The study reveals the profound moral conflicts of “dual identity” and “dual allegiance,” where nurses are torn between their roles as caregivers and as family members. This moral tension, often experienced as an ongoing inner struggle, adds to a unique emotional burden not adequately captured in existing literature.

By employing a hermeneutic phenomenological approach, the research gives voice to the lived experiences of nurses in high-risk regions, illuminating how they cope with ethical distress, fear, uncertainty, and responsibility. Their narratives highlight the centrality of virtues of the nursing profession such as courage, compassion, respect, honesty, and practical wisdom, demonstrating how **phronesis** guides real-time moral judgment when external guidelines are absent and ethical dilemmas emerge suddenly amid physical danger. The findings also underscore the importance of peer support, often valued more than family support in these contexts, and call for organizational policies, education and research that directly address the moral and emotional needs of nurses under extremely high-risk conditions and application to virtue ethics.

Importantly, the study shows that many commonly proposed organizational interventions—such as expanded services or on-site childcare—are important and necessary during active missile fire. Support must therefore be grounded in what is **operationally feasible** during an acute threat. Practical measures include consistent real-time communication, clear guidance on when to delay travel or shelter in place, flexible scheduling to reduce commuting during high-risk periods, and rapid staffing coordination based on who can safely arrive for work. During the crisis, many departments created on-site sleeping arrangements so participants could remain in the hospital and avoid dangerous travel after their shifts. These adaptive, context-sensitive responses required leaders to be flexible, creative, and present as conditions shifted from moment to moment.

Ultimately, this study offers a much-needed framework for understanding the complex realities faced by nurses in regions experiencing both health crises and armed conflict. The study extends theoretical work on moral distress, virtue ethics, and phronesis by illustrating how practical wisdom operates under conditions where external guidelines are absent, and where ethical decisions are made in real time amid physical danger. The hermeneutic phenomenological perspective further highlights that moral distress arises not only from system constraints but from disruptions in participants lived worlds with the merging of personal vulnerability, family responsibility, and professional duty. It informs leadership practices, crisis preparedness, and support systems, ensuring that frontline workers receive the recognition, resources, and ethical consideration they deserve. By centering the voices of nurses, this research contributes meaningfully to scholarly discourse and offers tangible implications for improving healthcare delivery and workforce resilience in high-risk settings.

Future research is planned to use the results of this study to develop a cross-sectional survey of nurses throughout the country to explore the themes uncovered in this research. Understanding broader implications, from a larger group of health professionals, using virtue ethics on the nursing population will provide insight into quality of life, stress-reducers, resilience and social support of those experiences in armed conflict.

## Figures and Tables

**Table 1 nursrep-16-00012-t001:** Demographics.

Participant	Age	Gender	Marital Status	Children	Degree	Unit	Nursing (Years)
P 1	42	female	married	3	BA	Internal Medicine	23
P 2	38	female	married	4	BA	Internal Medicine	14
P 3	39	female	divorce	2	BA	Internal Medicine	13
P 4	33	female	married	2	MA student	Intensive care	4
P 5	41	female	divorce	2	MA	Neonatal intensive care	7
P 6	38	female	married	7	BA	Oncology	12
P 7	43	female	married	2	MA	Internal Medicine	9
P 8	33	male	married	preg	MA student	Intensive care unit	6
P 9	44	male	married	1	BA	Neuro- surgery	4
P 10	44	female	married	4	Geriatric NP	Intensive care	20

**Table 2 nursrep-16-00012-t002:** Questions for the interview.

In what ways would you describe yourself as a person and as a nurse? In what ways do you cope with stress? How do you stay motivated to work every day? What gives you the strength to do so? How do you perceive your strengths, what gives you a sense of strength? What was the extent of the assistance you received from the organization?

## Data Availability

The qualitative data supporting the findings of this study cannot be publicly shared due to ethical and consent-related restrictions. De-identified excerpts (e.g., anonymized quotes or coding summaries) are available from the corresponding author upon reasonable request and contingent on institutional approval, depending on the requestor’s intended use.

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
