# Peer review of "Torn Between Identities: A Hermeneutic Phenomenological Study of Nurses’ Dual Allegiance During COVID-19 and Armed Conflict"

_nursrep, 2025, doi:10.3390/nursrep16010012_

Round 1
Reviewer 1 Report
Comments and Suggestions for Authors
The paper addresses a topic of exceptional relevance, giving voice to nurses who worked under the simultaneous pressures of the COVID-19 pandemic and armed conflict. The manuscript is well written, and participants’ narratives are presented with authenticity and respect, making a valuable contribution by highlighting the ethical and emotional complexity of professional identity in extreme conditions. Nevertheless, before it can be considered for publication, several areas require revision to ensure conceptual coherence, methodological transparency, and interpretive depth.
Introductory note on methodology
Several of the comments below depend on clarifying the chosen qualitative approach.
The manuscript currently blends elements of Interpretative Phenomenological Analysis (IPA) and hermeneutic phenomenology, which, although connected, stem from different philosophical traditions and analytic logics. Clarifying which framework guided the study (psychological-idiographic, as in IPA, or philosophical-existential, as in hermeneutic phenomenology) will help readers understand the rationale for your analytic choices and strengthen the methodological integrity of the paper. Once this clarification is made, the rest of the revisions will align naturally.
Methodological coherence and transparency
One important methodological point concerns the combination of hermeneutic phenomenology and Interpretative Phenomenological Analysis (IPA) described in your paper.
Both are interpretive traditions, but they serve different purposes:
- Hermeneutic phenomenology (Heidegger, Gadamer, van Manen) is a philosophical approach that explores the meaning of being and seeks to interpret lived experience in its existential depth.
- IPA (Smith et al., 2009) is a psychological and analytic method that operationalizes interpretation through systematic steps (reading, noting, identifying themes, moving between cases, and developing superordinate themes).
These two approaches can coexist successfully, but only when the relationship between them is made clear, that is, when hermeneutic phenomenology is presented as the philosophical lens, and IPA as the method of analysis used to operationalize interpretation.
At the moment, the manuscript seems to declare both approaches but not to demonstrate how they were integrated. For example, in the Methods section, you state that:
“We used a focused interpretative (hermeneutic) phenomenology approach… to understand the lived experience of nurses… Data were analyzed line-by-line and coded into categories.”
This description reads more like thematic or content analysis, because it focuses on coding and categorizing data, but does not show the interpretive movement typical of hermeneutic inquiry or IPA (e.g., moving between the parts and the whole, reflecting on pre-understandings, interpreting the participants’ meaning-making).
Similarly, in the Findings section, powerful quotations are presented — for instance:
“Leaving them (the family) in the sirens... making sure the children are safe... and not knowing how my husband was dealing with them.”
However, the commentary that follows mainly repeats or paraphrases what the participant said (“Nurses faced the difficult reality of leaving their families during dangerous situations...”) instead of interpreting what this experience reveals about being a nurse caught between professional duty and personal vulnerability, which would be the goal of a hermeneutic or IPA interpretation.
To strengthen methodological coherence, I recommend:
-
Clarify your epistemological stance: for example, you could write
“This study was guided by a hermeneutic phenomenological perspective inspired by Heidegger and Gadamer, operationalized through the procedures of Interpretative Phenomenological Analysis (IPA) described by Smith et al. (2009).”
This makes clear that hermeneutics provides the philosophical foundation, while IPA provides the analytic process. -
Explain the interpretive process: briefly describe how you moved from raw data to meaning. For instance:
“During analysis, we engaged in a hermeneutic dialogue with the data, moving iteratively between individual transcripts and the whole dataset. Each researcher reflected on personal pre-understandings and interpretations were refined through discussion.”
-
Make the results more interpretive: after each quotation, go beyond description to discuss the meaning. For example:
Instead of:“This quote shows that nurses were anxious about leaving their children.”
You could write:
“This quote illustrates how nurses experienced a moral fracture between care for others and self-preservation, an existential conflict that reshaped their sense of professional identity.” -
Use your voice as interpreter: In hermeneutic work, the researcher’s understanding is part of the meaning-making process. You might add a brief reflective note showing how your interpretation evolved.
The integration of hermeneutic phenomenology and IPA is not a mistake, but at present, it remains unclear and underdeveloped. Strengthening the connection between the philosophical framework and the analytic procedures would make your study far more coherent and persuasive. Your data are rich and meaningful; they already contain the essence of a truly hermeneutic interpretation. The next step is to make that interpretive reasoning visible to readers.
Data Collection
The Data Collection section provides useful information about the timing and mode of interviews; however, several key elements could be clarified or expanded to strengthen methodological transparency and ensure that readers fully understand how data were generated. The Data Collection section would benefit from a few additional sentences clarifying these aspects. These refinements would make the process more transparent and align the paper with international standards for qualitative reporting (e.g., COREQ). Your dataset appears rich and meaningful — adding these details will help readers appreciate the robustness and ethical integrity of your work. Please consider the following suggestions:
- You mention two data collection periods (April–May 2021 and December 2022–January 2023). It is not entirely clear whether these represent two distinct phases involving different participants or a continuation of the same sample.
Clarifying this point would help readers understand whether your study adopted a longitudinal design or a single cross-sectional data collection extended over time.
For example, specify whether the same nurses were re-interviewed or whether new participants were included in the second phase. - The paper includes the interview questions but does not explain how they were developed. In qualitative research, and particularly in phenomenological work, it is important to show whether the guide was informed by previous literature, theory, or pilot testing. You might add a sentence indicating how the questions were derived (e.g., from literature on moral distress or from the study’s conceptual framework) and whether any adjustments were made after initial interviews.
- It would be helpful to elaborate on who conducted the interviews and their relationship with the participants. For instance, was the principal investigator a colleague or supervisor in the same institution? Explaining how potential power dynamics were managed and how reflexivity was maintained (e.g., through a reflexive journal or team debriefing) would enhance the credibility of the study.
- Since interviews were conducted via Zoom, please provide a short reflection on how the virtual environment may have affected the interaction and data quality — for example, how privacy and confidentiality were ensured, or whether any limitations (technical issues, loss of non-verbal cues) were encountered. This is especially relevant given the emotional and ethically sensitive nature of your topic.
- You note that interviews were conducted in Hebrew and translated into English, but it remains unclear whether the analysis was carried out in the original language or in translation. It would strengthen rigor to describe how the accuracy of translation was checked, for example, whether key passages were cross-verified by bilingual researchers or back-translated to preserve meaning.
- Currently, you state that data collection continued until thematic saturation was reached. To enhance transparency, please explain how this was determined. In phenomenological research, saturation (or interpretive completeness) refers not only to repetition of themes but to the point where no new meanings emerge and understanding becomes sufficiently rich. A brief explanation, such as “After the ninth and tenth interviews, no new themes emerged and existing ones were consistently confirmed”, would be sufficient.
- It would be useful to describe how verbal consent was documented and securely stored (for instance, recorded at the start of each Zoom interview), and how participants’ privacy was safeguarded during online data collection. Given the sensitive context of war and moral conflict, these details help demonstrate ethical rigor.
- If you held regular discussions between researchers during data collection to reflect on emerging themes and emotional impact, it would be valuable to mention this. Such debriefing sessions are an important element of qualitative rigor and reflexive practice, particularly in studies dealing with emotionally charged narratives.
Data Analysis
The Data Analysis section would benefit from greater methodological transparency and closer alignment with the interpretive approach you have declared (hermeneutic phenomenology or IPA). The description provided (“coded into meaning units and categorized into subthemes and themes”) appears rather generic and does not sufficiently convey how you interpreted the lived meaning of participants’ experiences.
Below, provided detailed guidance to help you refine this section.
- You state that a “phenomenological–hermeneutic method” was used, but you do not specify which particular model guided your analysis (e.g., Heidegger/Gadamer, van Manen, Lindseth & Norberg, or the IPA model by Smith et al., 2009). Since these approaches differ in both philosophy and procedure, readers need to understand which one you followed. If you were inspired by IPA, you could specify that the analysis followed the stages proposed by Smith, Flowers, and Larkin (2009):
“reading and re-reading, initial noting, development of emergent themes, clustering of themes, and cross-case synthesis.”
If instead you drew on hermeneutic phenomenology, explain the interpretive process using its own logic — for example:
“analysis was guided by the hermeneutic circle, moving iteratively between individual narratives and the overall text, and by reflecting on pre-understandings to reach interpretive insight.”
This clarification will ensure philosophical coherence between your declared approach and your analytic procedures.
- Currently, the section emphasizes coding and categorization, which are descriptive steps, but does not show the interpretive reasoning typical of hermeneutic analysis.
In phenomenology, readers expect to see how you moved from description to interpretation, how you went beyond what participants said to reveal what their words mean. For instance, instead of only stating that transcripts were “divided into meaning units,” you could write:
“Each transcript was read several times to identify significant statements that captured the essence of the experience. These statements were then reflected upon in relation to the researcher’s pre-understandings and to the evolving interpretive framework, allowing latent meanings to emerge.”
Including one or two sentences like this would clearly show that your analysis was interpretive rather than merely descriptive.
- You mention that both authors discussed the data “until agreement was reached,” but more detail on this process would strengthen the study’s credibility. For example: Did each author analyze the transcripts independently and then meet to compare interpretations How were disagreements or differing perspectives resolved? Was reflexivity used to acknowledge each researcher’s position in relation to the data (e.g., insider vs. outsider)?
Adding a brief paragraph on this collaborative process — even just a few lines — would demonstrate analytic rigor and transparency.
- Because your interviews were conducted in Hebrew and translated into English, it would be important to explain whether the analysis occurred in the original language or after translation. Language carries cultural meaning, and in phenomenological research, this can influence interpretation. For example, you might add:
“Data were analyzed in Hebrew to preserve linguistic and cultural nuances; translated excerpts were used solely for publication. The research team verified translations against the original transcripts to ensure conceptual accuracy.”
Such a statement reassures readers that translation did not compromise interpretive depth.
- Hermeneutic analysis assumes that the researcher’s perspective is not a source of bias but a tool for understanding. Currently, the manuscript does not show how your pre-understandings shaped or interacted with the data.
You could add a sentence such as:
“Throughout the analysis, researchers engaged in reflexive discussions to examine how their professional backgrounds and emotional responses influenced interpretation.”
This would demonstrate methodological awareness and strengthen the interpretive integrity of your findings.
- Finally, to reinforce scholarly credibility, you might cite one or two methodological references consistent with your chosen approach. For example: If IPA: Smith, J. A., Flowers, P., & Larkin, M. (2009). Interpretative phenomenological analysis: Theory, method, and research. If hermeneutic phenomenology: van Manen, M. (1990). Researching lived experience or Lindseth & Norberg (2004).
These references will situate your method within a recognized analytic tradition.
The data you present are rich and emotionally powerful, but the analytic procedure currently reads as a descriptive thematic summary.
Clarifying the analytic framework, detailing how interpretation unfolded, and adding one short example would substantially enhance transparency and ensure full alignment between your philosophical stance and analytic execution.
This improvement will help readers appreciate the interpretive rigor underlying your insightful findings.
Rigor and trustworthiness
The section on Rigor and Reflexivity identifies several important strategies (member checking, triangulation, audit trail, and reflexive dialogue), but their application is only briefly mentioned and would benefit from further clarification and referencing. At present, the reader cannot clearly see how these strategies were enacted during the research process or how they contributed to ensuring the trustworthiness of the findings. Specifically:
-
Member checking: You state that this strategy was used, but it is unclear when and how participants were involved. Did they review summaries of their interviews, preliminary themes, or the final interpretation? Describing this briefly would demonstrate how participants’ feedback validated or refined your analysis.
-
Triangulation: It is not explicit what type of triangulation was applied (e.g., between researchers, data sources, or methods). Since two researchers were involved, it seems this was mainly investigator triangulation; clarifying this would make your claim of rigor more concrete.
-
Audit trail: You mention its use, but do not explain what documentation was included (e.g., analytic memos, meeting notes, theme development logs). Briefly indicating how this material was used to trace analytic decisions would enhance dependability.
-
Reflexivity: The statement that reflexive discussions were conducted “to maintain objectivity” is epistemologically inconsistent with interpretive phenomenology, where the aim is not objectivity but reflexive awareness. It would strengthen your paper to explain how reflexivity was practiced (for example, through reflexive journaling, analytic memos, or team debriefing sessions) and how it informed interpretation.
Overall, I recommend expanding this section to include short, study-specific examples rather than abstract definitions, showing what you actually did to ensure credibility, dependability, and confirmability, also referring to bracketing.
Finally, please consider including key references to support your claims of methodological rigor.
Results and interpretive depth
The results section contains rich and moving data, but it often reads as a descriptive summary of what participants said. To enhance interpretive rigor, each theme should include a brief interpretive synthesis explaining what the theme reveals about the moral and existential experience of nurses.
For example, when quoting participants who felt “torn” between duty and family, interpret this not only as stress or conflict but as a deeper negotiation of moral identity and selfhood.
This shift from description to interpretation will make your findings more aligned with a hermeneutic or IPA orientation. Also consider merging overlapping themes (e.g., “Personal Characteristics” and “Personal Conduct”) to improve focus and avoid redundancy.
Discussion and theoretical integration
The discussion successfully connects the findings to virtue ethics and moral distress, but it could go further in explaining how your study extends current knowledge.
-
Clarify why the dual-crisis context (pandemic + war) reveals new aspects of moral distress not previously explored.
-
Strengthen the link between your findings and virtue ethics by showing how specific virtues (courage, compassion, honesty, wisdom) emerged from participants’ experiences rather than being imposed after analysis.
-
Incorporate more recent international literature (2023–2024) on moral resilience and ethical practice in crisis settings to situate your work in a global context.
Reducing repetition between the Results and Discussion sections will also help maintain focus and improve readability.
Structure and balance
The Findings section currently occupies most of the paper, while the Discussion is relatively brief. A better balance between data presentation and interpretation will enhance the narrative flow.
Ensure that the paper follows a clear logic: context → methods → lived experiences → interpretive synthesis → (eventually) theoretical implications.
Ethical transparency and AI use
Your statement about the use of generative AI is appreciated and transparent. However, please specify the extent and purpose of AI assistance. It is important to confirm explicitly that no identifiable participant data were processed using AI and that all outputs were critically verified by the authors. This clarification aligns with MDPI’s policy on responsible use of generative tools.
Implications and conclusion
The conclusion effectively summarizes the key findings but would benefit from a clear paragraph outlining practical implications.
You might structure it as follows:
-
For nursing leadership: develop supportive structures for ethical reflection and peer dialogue during crises.
-
For education: integrate moral resilience and virtue ethics into training curricula.
-
For research: encourage cross-national comparisons and longitudinal studies on nurses’ moral experiences.
Adding this structure will make your conclusions stronger and more actionable for readers and policymakers.
Editorial and reference issues
Please review the manuscript for small typographical and formatting inconsistencies (spacing, capitalization, citation style). References should follow the Nursing Reports (MDPI) format consistently. Some recent or international sources on hermeneutic phenomenology and moral distress could further strengthen the reference list.
In summary, this is a thoughtful and important paper that addresses a meaningful phenomenon within nursing ethics and crisis care. With improved methodological alignment, enhanced interpretive analysis, and clearer implications for nursing practice, the manuscript could make a valuable contribution to the journal’s readership.
I hope these comments help you strengthen the paper for resubmission.
Author Response
Please see attached file.
Thank you for your review.

Reviewer 2 Report
Comments and Suggestions for Authors
The topic addressed is highly relevant, and the work presents a rich phenomenological description of nurses' experiences during pandemics and armed conflicts. The hermeneutic approach is consistent with the research objective, and the manuscript includes a wealth of valuable testimonies that enrich our understanding of the phenomenon.
However, substantial revisions are needed to improve rigor, clarity, and overall structure:
1. Introduction and Background
The introduction is very long, redundant, and includes a lot of descriptive information on the geopolitical context; I suggest streamlining it and focusing more on the scientific gap. Some passages repeat existing concepts (stress, resilience, dual crises).
The background can be reorganized to clearly distinguish between pandemic crisis, nursing experience in war contexts, and the lack of studies on the overlap between the two conditions.
2. Methods
The methodological section requires greater rigor and detail, in particular:
clarifying the sampling process (inclusion, exclusion, contradictory criteria: "community-based?" but hospital participants);
Clarify the consistency between dates (2021 and 2022–2023 interviews: rationale for the double collection?);
Specify how credibility and reflexivity were ensured with concrete examples;
Detail the thematic analysis process: steps, phases, coding, triangulation.
Some methodological claims (audit trail, repeated member checking, prolonged engagement for "over a year") must be supported by a concrete description.
3. Results
The results section is very extensive, narrative, and includes an excessive number of direct quotes.
I suggest synthesizing and integrating the quotes, retaining the most significant ones, clearly highlighting how the themes emerge (aggregation process), and avoiding repetitions across the seven categories, which present conceptual overlaps.
4. Discussion
The discussion is interesting but contains several points that repeat the results without critically analyzing them.
I suggest distinguishing what emerges as new with respect to the literature, reducing the references already widely discussed in the results, and better clarifying how ethical virtues and phenomenological theory enrich the interpretation.
Author Response

(The authors gave the same response as above.)

Reviewer 3 Report
Comments and Suggestions for Authors
See the attached PDF document.

Author Response

(The authors gave the same response as above.)

Reviewer 4 Report
Comments and Suggestions for Authors
Thank you for your research! My comments to improve your manuscript are:
- A paragraph should be devoted in the Introduction to explaining the phrase "focused interpretive hermeneutic, phenomenological approach". It is not clear how interviews implement this method.
- It is not obvious which scientific gap the research attempts to cover. Research questions and assumptions could answer this purpose.
- The background should focus on interviews as data collection methods. An independent Literature Review Section should be added to stress similar studies.
- Results should be discussed in light of research assumptions.
- The benefits of the study to nurses should be accentuated in the conclusion, clarifying how nurses can use the findings to better complete their duties. The same holds for other medical stakeholders, pointing out how they can use the results to relieve nurses.
- Finally, the extent of the results' generalization and the research future expansion should be described in the Conclusion.
Author Response

(The authors gave the same response as above.)

Round 2
Reviewer 2 Report
Comments and Suggestions for Authors
Dear authors,
thank you for the opportunity to revise your manuscript. However there are some modifications to do before consider this manuscript for publications:
I suggest making the use of the concepts of dual identity, dual allegiance, and moral conflict more consistent, avoiding potential conceptual overlap and clarifying any differences in meaning.
Although the section on reflexivity is present, it could be further strengthened by more clearly explaining how the professional role of researchers (particularly in leadership positions) has been managed to limit potential interpretive bias.
The discussion is well-structured, but in some places it is very dense. I suggest a reduction in conceptual repetition improving the flow of the text without compromising its interpretative depth.
It is recommended that the limitations related to the transferability of the findings be more clearly reiterated, consistent with the qualitative and contextual nature of the study.
Author Response
We are grateful for the revierwers insights and belive that the revisions have meaningfully strenghened the mauscript.

Reviewer 3 Report
Comments and Suggestions for Authors
After carefully examining the authors’ response letter and the explanations provided for each reviewer comment, I consider that the authors have addressed the vast majority of the suggestions in a thorough, transparent, and constructive manner. Overall, the revisions demonstrate a clear effort to strengthen methodological rigor, conceptual clarity, and alignment between findings, theory, and implications.
Conceptual framing and research focus
The authors adequately clarified that no pre-existing theoretical framework guided the research questions and that virtue ethics emerged inductively from the data analysis. This clarification resolves the initial concern regarding conceptual positioning and strengthens epistemological coherence. In addition, the knowledge gap has been more explicitly articulated, and the research questions are now more clearly answered in the Discussion, improving the manuscript’s internal logic and contribution.
Methodology, rigor, and transparency
The revisions substantially improve methodological transparency. Sampling procedures, recruitment pathways, inclusion criteria, and the absence of incentives are now clearly described. The addition of a dedicated section on rigor and reflexivity—including insider positioning, saturation criteria, member checking, triangulation, and bilingual verification—brings the manuscript into strong alignment with COREQ standards expected by Nursing Reports. These changes significantly enhance credibility, dependability, and trustworthiness.
Data analysis and presentation
The authors expanded the description of analytic procedures, clarified saturation, and strengthened the explanation of how themes were derived. Tables 1 and 2 have been reformatted and improved, and participant quotations are now consistently identified, allowing for better contextual interpretation. While the audit trail could always be further elaborated, the current level of detail is appropriate for publication in this journal.
Discussion, implications, and limitations
The Discussion has been meaningfully sharpened. The authors now explicitly link findings to the research questions, avoid overgeneralization, and clarify how the dual-crisis context contributes novel insights into moral distress. Theoretical, practical, and policy implications are more concrete and relevant to nursing leadership and organizational practice. Importantly, limitations and future research directions are now clearly stated, demonstrating reflexivity and scholarly maturity.
Remaining minor issues
Any remaining issues are minor and primarily editorial in nature (e.g., final stylistic polishing or journal-specific formatting checks). These do not undermine the scientific quality or coherence of the manuscript.
Author Response

(The authors gave the same response as above.)
